# The Influence of Symbiosis on the Proteome of the *Exaiptasia* Endosymbiont *Breviolum minutum*

**DOI:** 10.3390/microorganisms11020292

**Published:** 2023-01-22

**Authors:** Amirhossein Gheitanchi Mashini, Clinton A. Oakley, Sandeep S. Beepat, Lifeng Peng, Arthur R. Grossman, Virginia M. Weis, Simon K. Davy

**Affiliations:** 1School of Biological Sciences, Victoria University of Wellington, Wellington 6140, New Zealand; 2Department of Biological Sciences, University of Alberta, Edmonton, AB T6G 2E9, Canada; 3Centre for Biodiscovery, Victoria University of Wellington, Wellington 6140, New Zealand; 4Department of Plant Biology, The Carnegie Institution for Science, Stanford, CA 94305, USA; 5Department of Integrative Biology, Oregon State University, Corvallis, OR 97331, USA

**Keywords:** *Breviolum minutum*, proteomics, symbiosis, free-living, Aiptasia

## Abstract

The cellular mechanisms responsible for the regulation of nutrient exchange, immune response, and symbiont population growth in the cnidarian–dinoflagellate symbiosis are poorly resolved. Here, we employed liquid chromatography–mass spectrometry to elucidate proteomic changes associated with symbiosis in *Breviolum minutum*, a native symbiont of the sea anemone *Exaiptasia diaphana* (‘Aiptasia’). We manipulated nutrients available to the algae in culture and to the holobiont *in hospite* (i.e., in symbiosis) and then monitored the impacts of our treatments on host–endosymbiont interactions. Both the symbiotic and nutritional states had significant impacts on the *B. minutum* proteome. *B. minutum in hospite* showed an increased abundance of proteins involved in phosphoinositol metabolism (e.g., glycerophosphoinositol permease 1 and phosphatidylinositol phosphatase) relative to the free-living alga, potentially reflecting inter-partner signalling that promotes the stability of the symbiosis. Proteins potentially involved in concentrating and fixing inorganic carbon (e.g., carbonic anhydrase, V-type ATPase) and in the assimilation of nitrogen (e.g., glutamine synthase) were more abundant in free-living *B. minutum* than *in hospite*, possibly due to host-facilitated access to inorganic carbon and nitrogen limitation by the host when *in hospite*. Photosystem proteins increased in abundance at high nutrient levels irrespective of the symbiotic state, as did proteins involved in antioxidant defences (e.g., superoxide dismutase, glutathione s-transferase). Proteins involved in iron metabolism were also affected by the nutritional state, with an increased iron demand and uptake under low nutrient treatments. These results detail the changes in symbiont physiology in response to the host microenvironment and nutrient availability and indicate potential symbiont-driven mechanisms that regulate the cnidarian–dinoflagellate symbiosis.

## 1. Introduction

Reef ecosystems are formed by an obligatory endosymbiotic relationship between cnidarians and dinoflagellate symbionts from the family Symbiodiniaceae. This symbiotic mutualism is hugely important in near-shore oligotrophic seas, where corals are major primary producers and provide a three-dimensional landscape to promote the abundance and diversity of other species [1]. Nutrient exchange is integral to the cnidarian–dinoflagellate association, with the host being responsible for providing shelter and essential nutrients, such as inorganic carbon, nitrogen, and phosphorus, to the algal symbionts. In return, the symbionts support host energy requirements by providing photosynthetic products [2,3]. Glucose is the dominant form in which photosynthetically-fixed carbon is translocated to the host [4,5]. Essential amino acids, fatty acids such as palmitic, stearic and oleic acids, and other lipids such as wax esters, triacylglycerol and sterols, are also translocated [6,7].

The Symbiodiniaceae are physiologically and ecologically diverse, with eleven genera described to date [8,9,10,11]. When *in hospite*, the symbionts are located in a host-derived late arrested phagosome known as the symbiosome [12,13]. The physiology, morphology, and cell cycle of the symbiotic dinoflagellates have long been known to change in response to the symbiotic state (i.e., *in hospite* vs. *ex hospite*) [14,15]. For example, when in symbiosis, the symbionts exist in a coccoid form, lacking flagella and having thinner cell walls, whereas the free-living dinoflagellates revert to a motile, gymnodinioid form with two flagella [13,14,15,16]. Moreover, in symbiosis the dinoflagellates release a substantial portion of their photosynthate to the host, whereas they typically release only a small fraction when free-living [17,18]. However, our understanding of the cellular and molecular mechanisms that underlie the establishment, maintenance and collapse of the cnidarian–dinoflagellate symbiosis remains limited [19], and researchers in the field have focused more on events occurring in the cnidarian rather than in the dinoflagellate partner [20,21,22,23,24].

In recent years, there has been an effort to remedy this disparity, with transcriptomic, proteomic, and lipidomic analyses revealing changes in transport components, photosynthesis, the lipid metabolism, cell division, and signalling when the alga is shifted to/away from a symbiotic lifestyle [25,26,27,28]. Proteins represent the catalytic (transporters, motors, enzymes), regulatory, and structural machinery by which organisms respond to their changing environment, and unlike transcripts, the identification of ‘symbiosis associated proteins’ and their biochemical characteristics provides new insights into cellular adjustments that occur as the algae transition between growth as a symbiont and free-living organism. Therefore, proteomics, which encompasses the study of protein identification, quantification, and interactions in cells or among organisms under defined conditions, is a powerful approach for elucidating how marine organisms acclimate to changes in their environment [29,30].

Here, we used proteomic methods [31] to further determine the changes associated with the establishment of the cnidarian–dinoflagellate symbiosis, concentrating on the effects of symbiosis and its accompanying nutrient microenvironment on the dinoflagellate endosymbiont [25]. Specifically, proteomic profiles of *Breviolum minutum*, a native symbiont of the model cnidarian *Exaiptasia diaphana* (commonly referred to as ‘Aiptasia’) [32,33], were compared between endosymbionts grown *in hospite* and *ex hospite*. In addition, comparisons of the different growth modes of the alga were made under a range of nutrient conditions/feeding regimes to take nutrient availability into account, which is likely to be very different for algal cultures in enriched seawater medium compared to *in hospite*, where the nitrogen supply (and perhaps other nutrients) is thought to be actively restricted by the host as a means of limiting symbiont growth [34,35,36]. Comparative proteomics was used to identify algal proteins and cellular pathways responsible for the proliferation, maintenance and fitness of the dinoflagellate symbiont in the endosymbiotic state, which are important for understanding a tightly-integrated and ecologically important symbiosis based on nutrient exchange.

## 2. Materials and Methods

### 2.1. Experimental Organisms and Design

*Breviolum minutum* (strain FLAp2, originally from Long Key, Florida) was cultured in Guillard’s f/2 medium (AusAqua Pty, SA, Australia) in artificial seawater (ASW) (Coral Pro Salt, Red Sea, New Zealand); this medium has a much greater nitrogen concentration than ASW, with nitrate concentrations of 883 µM and 30 µM, respectively [37]. Specimens of Aiptasia were taken from a clonal laboratory stock (n = 200; clonal strain ID: NZ1). Strain NZ1 is originally from the Indo-Pacific region and naturally and exclusively hosts *B. minutum*, although a culture of *B. minutum* from Aiptasia NZ1 was not available at the time of this experiment.

Algal identity, both in culture and in the anemones, was confirmed by sequencing the internal transcribed spacer (ITS2) region of the ribosomal DNA. Briefly, DNA was extracted using CTAB/phenol-chloroform [38], and the ITS2 DNA amplified by PCR using ITSintfor2 (5′-GAATTGCAGAACTCCGTG-3’) and ITS2Rev2 (5′-CCTCCGCTTACTTATATGCTT-3′) as the forward and reverse primers, respectively [39,40]. Amplicons were directly sequenced (Sanger sequencing; Macrogen Inc., Seoul, South Korea), sequences were aligned using Geneious Prime v. 2019.2.3 (Biomatters Ltd., Auckland, NZ), and a custom BLAST search was performed against Symbiodiniaceae ITS2 sequences in Geosymbio [41].

Anemones and cultured *B. minutum* were used immediately from laboratory stocks; pre-acclimatization was not performed to reduce mortality, especially of cultured algae maintained in ASW. *B. minutum* cultures were divided into fresh ASW and f/2 treatments (250 mL uncapped flask, n = 9 *per* treatment). Likewise, anemones were divided into “well-fed” and “starved” treatments (250 mL uncapped containers, n = 9 *per* treatment, 10 anemones *per* container). The well-fed anemone treatment was fed five times *per* week with *Artemia* sp. nauplii and the starved anemones were not fed during the experiment. All treatment media (i.e., ASW and f/2) were changed every day. The experiments were continued for 21 days.

All treatments were maintained under an irradiance of 100 µmol photons m^−2^ s^−1^ on 12 h:12 h light:dark cycle at 25 °C. Pulse amplitude modulated (PAM) fluorometry (Diving-PAM, Walz, Germany) was used to measure the maximum photochemical quantum yield of photosystem II (calculated as maximal variable fluorescence/maximal fluorescence, F_v_/F_m_ = (F_m_ − F_0_/F_m_) every three days, following a one-hour dark adaptation period at the end of the daily light cycle, to assess the photosynthetic health of the algae. The measurement distance was a standard 5 mm. All PAM settings were maintained over the course of the experiment: measuring light 4, saturation intensity 4, saturation width 0.6 s, gain 2, and damping 2. Mean (±SE) F_v_/F_m_ was calculated based on measurements from each sample (n = 9 *per* treatment) at each time-point.

At the end of the experiment (Day 21), anemones from each treatment were pooled (5–7 anemones per replicate, n = 9 replicates per treatment) and rapidly frozen at −80 °C. Anemones were homogenized using a tissue homogenizer at 4 °C, and symbionts were separated from the host by centrifugation at 500× *g* for 5 min at 4 °C. The host fraction was used for total protein quantification by fluorometry with a Qubit Protein Assay Kit (ThermoFisher Scientific). The symbiont pellet was resuspended and washed at 4 °C with 500 µL ASW, a 10 µL subsample was taken for quantification of cell number using a haemocytometer, and the remaining resuspension was pelleted again by centrifugation at 500× *g* for 5 min at 4 °C. Free-living algae (i.e., in f/2 and ASW) were also centrifuged at 500× *g* in 50 mL Falcon tubes at 4 °C and then snap-frozen at −80 °C. All samples were stored at −80 °C prior to analysis.

### 2.2. Protein Extraction

Algal pellets were washed with 500 µL cold HPLC-grade water at 4 °C to remove salts and resuspended in 500 μL 1% sodium deoxycholate (SDC) in HPLC grade water to help remove host tissue from the agal cells. This suspension was passed through a 23-gauge needle (0.337 mm inner diameter) five times to remove any remaining protein of host origin. All samples were centrifuged again at 500× *g* for 5 min at 4 °C and the host-containing supernatant was discarded. For lysis, the pellet was resuspended in 500 µL 5% sodium dodecyl sulphate in HPLC-grade water and homogenized with an ultrasonic probe (Vibra-Cell™ Ultrasonic VCX 500). Total protein extraction was conducted by incubating homogenized cell samples at 85 °C for 20 min with 1% β-mercaptoethanol. Hydrophobic pigments were removed by ethyl acetate phase transfer. A filter-aided sample preparation method was used to concentrate and purify the protein fraction, followed by trypsin digestion overnight. Digested peptides were separated using a 30 kDa molecular-weight cut-off filter, any remaining SDC was precipitated by the addition of formic acid to a final concentration of 1%, and the sample desalted with C18 tips (Omix, Agilent Technologies) and resuspended in 1% formic acid.

### 2.3. LC-ESI-MS/MS

Liquid chromatography-electrospray ionization-tandem mass spectrometry (LC-ESI-MS/MS) was used for peptide analysis. A total of 200 ng peptides per sample were separated by liquid chromatography (Ultimate 3000, Dionex) on a 150 mm PepMap C18 column (#160321, Thermo Scientific) at 35 °C with an 80 min linear gradient from 95:5–65:35% buffer A (0.1% formic acid):buffer B (80% acetonitrile, 0.1% formic acid) at a flow rate of 0.3 μL min^−1^. Peptides were ionised at 1.8 kV and analysed by an Orbitrap Fusion Lumos Tribrid mass spectrometer (Thermo Scientific) using Thermo Xcalibur (v4.3). Parent ion spectra were acquired using the Orbitrap with a resolution of 120,000, scan range of 375–1599 *m*/*z*, AGC target 5.0 × 10^3^, 50 ms maximum injection time, and allowed charge states 2–7. Sequencing was performed via higher energy collisional dissociation (HCD) fragmentation (collision energy 30%) on the top 20 abundant precursor ions, and the fragment ions spectra were acquired in the linear ion trap with the isolation window set at 1.6 m/z and dynamic exclusion enabled for 60 s. Mass spectrometry data are publicly available from the ProteomeXchange Consortium [42] via the PRIDE [43] repository with identifier PXD036981.

### 2.4. Protein Identification and Quantification

The Andromeda algorithm in MaxQuant v1.6.17.0 [44] was used to search the spectra against a custom protein database constructed from a *B. minutum* transcriptome [45] and a list of common protein contaminants. A maximum of two missed cleavages, a minimum of two matching peptides *per* protein, and a minimum peptide length of seven amino acids with a false discovery rate (FDR) of 0.01 were required for peptide and protein validations. Variable modifications were set as N-terminus carbamylation and oxidation of methionine, and fixed modifications were specified as cysteine carbamidomethylation. The ‘match between runs’ feature was enabled. All spectra from *in hospite* treatments were also searched against Aiptasia protein sequence databases [46] to verify that few contaminating host proteins were present.

### 2.5. Statistical Analyses

Cell counts and maximum quantum yield data were checked for normality by the Shapiro–Wilk test. A two-sample *t*-test was used to compare the densities of symbionts from *in hospite* treatments. Mixed two-way analysis of variance (ANOVA) was used to test for differences between the maximum quantum yields between different treatment groups and days, followed by a Bonferroni post hoc test to identify any pairwise differences. All physiological data analyses were conducted in R 4.0.3 [47].

Perseus v.1.6.13.0 [48] was used to remove known contaminant proteins and false identifications. The remaining protein precursor intensity was log_2_-normalized and further used to compare and identify significantly different proteins between treatments using polyStest [49]. The FDR and log-ratio thresholds were set to ≤0.05 and ±0.7, respectively. All detected proteins were searched against the UniProtKB database using DIAMOND [50] with an E-value ≤ 1 × 10^−5^. The top SwissProt (manually curated proteins) matches were used to assign protein sequence IDs and annotations, unless no SwissProt matches were detected, in which case TrEMBL matches were assigned. Any unmatched sequences were designated as hypothetical proteins. Principal component analyses were plotted by ClustVis, using ln(x) transformed protein abundance values and the singular value decomposition method [51].

## 3. Results and Discussion

### 3.1. Endosymbiont Density and Photobiology

ITS2 sequencing confirmed the genetic identity of the algae in all treatments as exclusively *B. minutum*. Symbiont cell densities of well-fed anemones were approximately 1.6-fold higher than starved anemones (Figure 1A). The F_v_/F_m_ of algae cultured in ASW and *in hospite* in starved anemones declined significantly (~38%) from Day 3 onwards, while algae in the nutrient-enriched f/2 and well-fed treatments showed no significant change in their F_v_/F_m_ (Figure 1B).

The host may have a reduced capacity to provide nitrogen or other nutrients (e.g., P) to the symbionts when starved due to either lack of supply or active restriction of resources by the host to the symbiont. This may subsequently result in nitrogen stress and reduced symbiont cell division [52], and may also impact the nitrogen available for photosystem protein synthesis and repair, potentially explaining the reduction in measured F_v_/F_m_ values.

### 3.2. Proteins Affected by Symbiotic State and Nutritional Regime

Symbiosis and nutritional state have been reported to have considerable impacts on the *B. minutum* proteome, suggesting a physiological response to the intracellular acidic environment of the symbiosome [53]. Changes of proteins observed in the current study included: (1) modifications to the cell wall; (2) responses to prevent host phagosome fusion with the lysosome; (3) responses to carbon, nitrogen and iron excess/limitation; (4) protective measures against oxidative stress; and (5) changes in carbohydrate and fatty acid synthesis and metabolism (Table 1 and Table 2).

A total of 2840 proteins were identified across all treatments, of which 56 known false matches and contaminant proteins were discarded. Principal component analysis plots suggest protein abundances as being distinctly different between symbiotic states (Figure 2A) and nutritional regimes (Figure 2B). A total of 1074 proteins were differentially expressed across all treatments (Appendix A). Based on homology, 1037 of these matched sequences in the UniProtKB database (E-value ≤ 1 × 10^−5^), and the remaining unmatched 37 sequences were designated hypothetical proteins. In other analyses, a total of 473 proteins were significantly different between the *ex hospite* (ASW and f/2 combined) and *in hospite* (well-fed and starved combined) states, and 684 proteins were differentially expressed between the high (well-fed and f/2) and low (starved and ASW) nutrient regimes (Appendix A). A total of 13 and 193 proteins were exclusively expressed in the symbiotic and free-living states, respectively. A total of 42 proteins were uniquely detected in both high nutrient regimes (well-fed and f/2), whereas only one protein was exclusively detected in both low nutrient regimes (starved and ASW).

### 3.3. Proteomic Shifts Associated with Symbiotic State

#### 3.3.1. Response to the Symbiosome

When *in hospite*, dinoflagellate symbionts live inside the symbiosome, which is a late arrested phagosome; it has been suggested that symbionts avoid host autophagy and immunodetection by manipulation of phosphoinositide (PI) metabolism pathways [54]. PI glycerolipids play an important role in signal transduction, cytoskeleton architecture, membrane dynamics and vesicle trafficking routes within eukaryotic cells [55]. The manipulation of PI metabolism pathways by PI kinases or phosphatases is the main mechanism for cell entry and preventing host autophagy by various microorganisms [55,56]. In particular, PI levels associated with phagosomes are modified by vacuolar pathogens to mimic subcellular compartments or arrest fusion with lysosomes [55]. Interestingly, a homologue of phosphatidylinositol phosphatase (INP53) was exclusively found in symbiotic state algae, raising the possibility that it functions by preventing phagosome maturation (Figure 3; Table 1). Moreover, a homologue of glycerophosphoinositol permease 1 (GIT1) was also only detected in the symbiotic state. GPI, a phospholipid metabolite produced by deacylation of PI, facilitates the transport of glycerophosphoinositol (GPI) across the cell membrane. GPI has been proposed to be a source of phosphate for cells experiencing mildly acidic or phosphate-limited conditions [57,58]. However, it is unknown whether INP53 and GPI work in concert in a functional cnidarian–dinoflagellate symbiosis (Figure 3). In addition, a homologue of Eukaryotic Elongation Factor 1 (eEF1), which is a multifunctional protein without any alternative forms [59], was uniquely expressed in symbiotic algal cells. The canonical function of eEF1 is in translation; however, these highly abundant and conserved proteins have also been reported to be involved in immunosuppressive processes of eukaryotic and prokaryotic pathogens [60]. An immunosuppressive function of eEF1 in cnidarian–dinoflagellate associations remains to be fully investigated.

Glycanosyltransferases (1,3-β-glucanosyltransferases; GAS) are GPI-anchored cell surface proteins that play a role in the elongation of β-1,3-glucan chains, which have a major role in cell wall assembly, hardening and softening [61]. The GAS family is essential for cell wall maturation, morphogenesis and virulence [61]. The dinoflagellate cell wall is typically thinner when in symbiosis, potentially facilitating the translocation of nutrients and communication between partners [13,62,63]. Consistent with this observation, two homologues of the GAS family were abundant in the current study; one (GAS2) was only found *in hospite* while the other (GAS4) was less abundant when *in hospite*, suggesting potential functions of these proteins in symbiont cell wall modification (e.g., softening or hardening) (Figure 3; Table 1).

#### 3.3.2. Dissolved Inorganic Carbon Transport

Photosynthesis is a critical feature of the cnidarian–dinoflagellate symbiosis, given the importance of metabolic exchange for this relationship. Symbionts can obtain inorganic carbon as CO_2_ for photosynthesis from holobiont respiration, calcification (in corals, where CO_2_ is produced as a by-product), or as dissolved inorganic carbon (DIC) in ambient seawater [19]. However, DIC in ambient seawater is mostly present as bicarbonate [HCO_3_^-^], and only CO_2_ can pass freely through cell membranes.

Host-derived carbonic anhydrase (CA), which catalyses the interconversions between inorganic carbon species, CO_2_, HCO_3_^-^ (bicarbonate) and CO_3_^−2^ (carbonate), is integral to cnidarian carbon concentrating mechanisms (CCMs), as it provides DIC in forms that can be used for both photosynthesis and calcification [64]. An increase in host CAs may be important for trafficking DIC to the symbiont where it can be fixed through the Calvin-Benson-Bassham cycle [20,65,66]. In the current study, four carbonic anhydrase homologues were significantly more abundant (>3-fold) in cultured *B. minutum* than in algae maintained *in hospite*, two of which were exclusively found in free-living cells (Table 1). Based on homology, some of these CAs are putatively localised to the mitochondria, but mitochondrial CAs have also been proposed to enhance photosynthesis by the recapture of respiratory CO_2_ [67]. Moreover, a homologue of V-type ATPase, a proton pump, was abundant in free-living algae irrespective of the nutritional state (3.21-fold). Proton pumps use ATP as a source of energy to create a proton gradient and function in harmony with carbonic anhydrases as part of the CCM [68,69]. Therefore, increased algal CAs and V-Type ATPase in the free-living state may be a response to a lack of readily available CO_2_ provided by the host CAs and proton pumps (Figure 3).

#### 3.3.3. Nitrogen Metabolism

Nitrogen flux is an integral part of the cnidarian–dinoflagellate symbiosis, especially in oligotrophic tropical waters [70,71,72]. Dinoflagellate symbionts assimilate the majority of nitrogen by the holobiont via the glutamine synthetase/glutamine 2-oxoglutarate amido transferase (GS/GOGAT) pathway, which generates glutamate as the final product, a precursor for other amino acids or purines. The dinoflagellate symbionts can also incorporate ammonium via glutamate dehydrogenase (GDH), which has a lower ammonium affinity. Nevertheless, the cnidarian host can also assimilate nitrogen [73,74,75,76]. Eleven different homologues of glutamine synthetase (GS), two of glutamate synthase 1 (GOGAT), and two of glutamate dehydrogenase 2 (GDH) were detected in the alga in this study. Four homologues of proteins with predicted functions in the GS/GOGAT pathway were differentially less abundant *in hospite* in comparison to the free-living state (Figure 3; Table 1). This lower abundance of GS/GOGAT in symbiosis is seemingly contradictory to the alga’s role in nitrogen assimilation; this may reflect either a relative reduction in algal demand for nitrogen due to simultaneous inorganic carbon limitation, or greater nitrogen availability in symbiosis due to the presence of host metabolic products despite host mechanisms to control the endosymbiont population by limiting nitrogen supply [18,35,77,78,79].

Dinoflagellate symbionts can also exclusively assimilate nitrate and nitrite by converting them to ammonium via specific reductase enzymes [80]; upregulation of nitrate and nitrite reductase transcripts has been reported from *B. minutum* when *in hospite* [26]. Nitrate transporter and nitrite reductase homologues were detected here; however, no differential abundance was detected between any of the treatments.

A nitrilase homologue was detected exclusively in the symbiotic state under both nutrient regimes and was particularly abundant in well-fed anemones. Nitrilases hydrolyse nitrile compounds to carboxylic acid and ammonia, and are found in a broad range of organisms, including bacteria, fungi and plants. Nitrilases are also involved in defence, detoxification and plant hormone synthesis [81]. Nitrilase has been reported to be less abundant during nitrogen starvation in marine green algae [82]. Our results may indicate that the symbionts respond to nitrogen starvation inside the host by hydrolysing nitrile compounds for ammonia (Figure 3).

With respect to the fate of nitrogen after assimilation, a group of 23 proteins were detected with predicted functions in methionine de novo synthesis and metabolism, of which 10 were more abundant in one or more of the different treatments (Appendix A). Methionine is required for protein synthesis and has been previously reported to be important in the cnidarian–dinoflagellate symbiosis, with both cnidarian host and dinoflagellate symbiont possessing methionine metabolic pathway enzymes [83]. Moreover, the native *B. minutum* has been hypothesized to support the host with greater amounts of methionine in comparison to non-native *Durusdinium trenchii* [20]. Unsurprisingly, given the links to protein synthesis, methionine pathway proteins in the current study were more abundant in both the nutrient-enriched (f/2) culture and well-fed anemone treatments than in the low nutrient/starved treatments.

Two important enzymes in proline biosynthesis from glutamate, gamma-glutamyl phosphate reductase (GPR) and pyrroline-5-carboxylate reductase (P5CR) [84,85], were only detectable in culture, irrespective of nutritional state. Proline biosynthesis and accumulation typically increase under different environmental stresses such as high salinity, UV radiation, heavy metals, oxidative stress and biotic stress [85], and also can function in osmotic control, as an antioxidant and in immune responses. Increased proline biosynthesis might therefore be indicative of a response to more stressful conditions when living outside of their natural habitat. The other way in which Symbiodiniaceae metabolize glutamine is via the purine pathway, which facilitates the storage of high-nitrogen compounds [75]. A homologue of guanosine monophosphate synthetase (GMPS), which functions in de novo synthesis of guanine nucleotides by hydrolysing glutamine [86], was more abundant in the free-living alga (1.16-fold) and especially in cultures grown in f/2 medium.

#### 3.3.4. Carbohydrate and Fatty Acid Metabolism

In total, 20 proteins involved in carbohydrate metabolism were differentially expressed, 17 of which were exclusively detected in the free-living state irrespective of the nutrient status (Appendix A). In contrast, homologues of ribulose-5-phosphate-3-epimerase and glucose-6-phosphate 1-epimerase, which are components of the oxidative pentose phosphate pathway, were more abundant in symbiosis. The pentose phosphate pathway, operating in parallel with the Embden-Meyerhof-Parnas glycolytic pathway, is fundamental to maintaining cellular carbon homoeostasis, moderating oxidative stress, and driving anabolic processes including the biosynthesis of nucleotides and amino acids [87,88].

Seven of the proteins involved in carbohydrate metabolism are alpha-amylase, polysaccharide monooxygenase, and 1,4-beta-D-glucan cellobiohydrolase B (CBHB), which hydrolyse polysaccharide chains to smaller sugars such as glucose (Appendix A) [89,90,91]. Interestingly, a specific homologue of an extracellular CBHB was more abundant in algae in the symbiotic state, although various extracellular cellulose and carbohydrate-binding proteins were more abundant in the free-living state. The function of these enzymes and the binding proteins are not known, but they could serve to tailor cell wall architecture, which in the Symbiodiniaceae is composed primarily of cellulose and glycoproteins [92] to the alga’s life style. Specifically, the algae would discard cellulosic thecal plates during ecdysis, where the plasma membrane and thecal vesicles are shed and new thecal vesicles are formed; this is well known to occur in these algae for which shed thecal plates can accumulate at the bottom of culture flasks and inside the symbiosome membrane [12,13,15]. Hence, these proteins may play a role either in the shedding process or in digesting shed cellulosic plates, whether inside the symbiosome or while free-living. The fate of digested polysaccharide complexes in the symbiosome remains to be elucidated, but it is noteworthy that an increased abundance of the algal sugar transporter SWEET1 was observed in cultures maintained in ASW (Appendix A), which might indicate that free-living algae scavenge fixed carbon from shed thecal plates and/or dead cells.

Lipids and fatty acids are the major form of energy storage in dinoflagellate symbionts and reflect the nutritional state of the holobiont [93,94]. Two homologues of acetyl-CoA carboxylase, the rate-limiting enzyme in fatty acid biosynthesis, and a group of other plastid-localised or cytosolic proteins involved in this pathway, were detected in all treatments. However, this group was significantly more abundant in the free-living algae (Appendix A). Different members of Symbiodiniaceae have different strategies for energy and fatty acid storage, which can be modified in symbiosis [3,95,96,97]. However, lower amounts of lipid and fatty acids have been detected in cultured Symbiodiniaceae in comparison to when *in hospite* [95]. It is unclear why proteins involved in lipid and fatty acid pathways were more abundant when *ex hospite* in the current study, although it could be due to release from host metabolic control.

Furthermore, four different homologues of polyketide synthase, such as phenolphthiocerol/phthiocerol polyketide synthase (PpsA) and highly reducing polyketide synthase 19 (PKS19), were exclusively found in free-living symbionts under both nutritional regimes. Polyketides, formed from acyl building blocks, are the backbone of some biological toxins. Polyketide metabolites in *Symbiodinium* sp., as well as polyketide synthetase genes from *B. minutum*, have been previously reported [98,99]. PpsAs have diverse biological functions, including having antifungal and antibacterial properties; however, in other symbiotic associations, these proteins are also involved in combating infection and increasing immunity [100,101]. Therefore, it is not clear if algal polyketide synthase is involved in the invasion of the cnidarian host by the symbiont, or whether it functions in synthesizing antibiotics for defending the holobiont against other potential endosymbionts or various microorganisms, such as those commonly associated with Symbiodiniaceae cultures [102].

### 3.4. Proteomic Shifts Associated with Nutritional State

#### 3.4.1. Photosynthesis

Photosynthesis is central to the cnidarian–dinoflagellate symbiosis (described in detail in Davy et al., 2012 [103]). A large number of proteins that have predicted functions in photosynthesis were identified from all the treatments in the current study. However, a group of photosystem (PS) I and II proteins involved in light harvesting, including chlorophyll *a*-chlorophyll *c*_2_-peridinin-protein complex (apcPC; mis-annotated in UniProt as “fucoxanthin-chlorophyll a-c- binding protein A/C”, see Jiang et al. 2014), chlorophyll *a-b* binding (LHC) and caroteno-chlorophyll *a-c*-binding proteins, as well as the chloroplastic electron transport chain-like cytochromes, ferredoxin-NADP reductases, P700 chlorophyll a apoproteins, PS I and II reaction centre proteins, including D1 and D2 proteins, became highly elevated in abundance in the nutrient-enriched treatments (Figure 3; Appendix A). In conjunction with this, proteins that function in different stages of the Calvin-Benson-Bassham cycle, such as ribulose-1,5-bisphosphate carboxylase/oxygenase (RuBisCO), fructose-1,6-bisphosphatase 1 (FBPase1), glyceraldehyde-3-phosphate dehydrogenase and phosphoribulokinase, were also most abundant in the high nutrient treatments (Appendix A). It was previously shown that the amount of chlorophyll per symbiont cell, photosynthetic efficiency and capacity, and Calvin-Benson-Bassham cycle activity all increase in a high-nutrient environment [104,105]. Conversely, low nutrient availability induces photoinhibition, reduced F_v_/F_m_ and chlorophyll content per cell [106,107], as was also observed in this current study (Figure 1B). Therefore, the upregulation of photosystems and Calvin-Benson-Bassham cycle proteins seen here is likely due to the increased availability of nutrients for protein synthesis by the dinoflagellate symbionts.

#### 3.4.2. Oxidative Stress under High Nutrient Regimes

Photosynthesis and respiration inevitably create reactive oxygen species (ROS) as by-products which can damage DNA, proteins and cellular membranes [108,109]. The importance of Symbiodiniaceae ROS generation and antioxidant activity in response to stressful conditions, such as increased temperature or irradiance, has been investigated before [109,110,111,112,113]. Superoxide anion radical (O_2_^−^) is produced by PS I and II, however it is reduced to hydrogen peroxide (H_2_O_2_) by superoxide dismutase (SOD) and further to H_2_O by ascorbate peroxidase (APX) or catalase [109]. Homologues of symbiont antioxidants, such as APX and SOD, were more abundant in the two high nutrient treatments (f/2 culture and well-fed *in hospite*). Moreover, four homologues of glutathione s-transferase and a homologue of chloroplastic glutathione reductase, all of which are important in the oxidative stress response and cell signalling, were more abundant in the high nutrient treatments. Increased gross photosynthesis in nutrient enriched Symbiodiniaceae cultures was previously reported [114]. Thus, the increased levels of proteins involved in symbiont antioxidant mechanisms in response to nutrient enrichment might be a cellular response for maintaining high rates of photosynthesis/respiration [115,116] (Figure 3; Table 2).

A homologue of metacaspase-1B (CASB) was also more abundant (2.35-fold) in the well-fed treatment, possibly due to stress induced by the high symbiont population density in the host’s tissues [117,118]. Oxidative stress can cause programmed cell death (apoptosis) in the Symbiodiniaceae [119,120]. Metacaspases, ancestors of metazoan caspases, are cysteine-type proteases [121]. Caspases from cnidarian–dinoflagellate associations function in symbiont expulsion from the host cell during stress [54,122]. More recently, metacaspase transcripts have been shown to increase in Symbiodiniaceae when *in hospite* as a response to thermal and pH stress [123,124]. Nevertheless, whether CASB activity is essential for cell cycle progression or is a response to oxidative stress-induced apoptosis remains to be demonstrated.

#### 3.4.3. Iron Metabolism

Iron is essential for photosynthetic microalgae because it is needed for many cellular processes and biochemical pathways, such as DNA synthesis, respiration and photosynthesis [125,126]. In Symbiodiniaceae, iron is a key metal in cell growth [127,128]. Moreover, iron uptake in Symbiodiniaceae (specifically *B. minutum*) increases at elevated iron concentrations, while iron utilization increases in efficiency in iron-depleted environments [129]. This is consistent with the findings from the current study, in which multicopper oxidase (MCO) was only detected under nutrient enrichment (i.e., f/2 and well-fed anemones) (Figure 3). MCOs are involved in the iron uptake pathway by oxidizing iron II to iron III in unicellular algae [130,131]. Interestingly, in another study, coral symbionts were identified as the main destination for heterotrophically-acquired iron [132]. This might help to explain the even greater abundance of MCOs in symbionts from heavily-fed anemones vs. their free-living counterparts in f/2 medium. In contrast, algae in the low nutrient regime (ASW and starved combined) increased expression of a homologue of soma ferritin (1.07-fold), an important iron storage protein (Figure 3). Economizing iron storage via ferritin in marine unicellular algae has been suggested as a common strategy for living in iron-depleted environments [133].

## 4. Conclusions

We identified proteins in *B. minutum* that are differentially abundant between the in and *ex hospite* states. The predicted functions of these proteins relate to the establishment and maintenance of a functional cnidarian–dinoflagellate symbiosis. Homologues of inositol permease and phosphatase, along with other immunosuppressant proteins, were only expressed *in hospite*, suggesting active manipulation of host cellular immune mechanisms by the native symbiont, possibly to arrest phagosome maturation. Additionally, homologues of carbonic anhydrases were expressed only when *ex hospite*, emphasizing the importance of DIC for algal growth and maintenance. Proteins with predicted functions in nitrogen assimilation were relatively less abundant *in hospite*; and a homologue of nitrilase with a predicted function in ammonium uptake from nitrile compounds was expressed exclusively *in hospite*. All of these observations are consistent with previous reports showing nitrogen limitation of the symbionts by the host [35,36,77]. Additionally, nutrient enrichment had a direct impact on proteins associated with photosynthetic electron transport, the Calvin-Benson-Bassham cycle, and iron metabolism. These data provide new target proteins and pathways in the symbionts for further studying of the mechanisms associated with the cnidarian–dinoflagellate symbiosis.

## Figures and Tables

**Figure 1 microorganisms-11-00292-f001:**
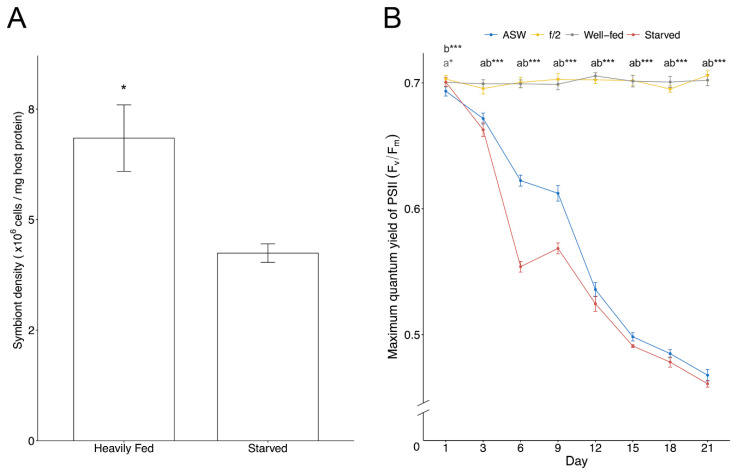
Physiological effects of different nutritional regimes in *Breviolum minutum* and Aiptasia. (**A**) Comparison of *in hospite* symbiont cell densities between well-fed and starved Aiptasia. n = 13 per treatment. (**B**) Maximum quantum yield of photosystem II (F_v_/F_m_) of cultured algae (ASW vs. f/2 medium) and *in hospite* (well-fed vs. starved). n = 10 per treatment at each time-point. Asterisks indicate significant differences: * *p* < 0.05. *** *p* < 0.0001. Different letters next to asterisks indicate significant differences between treatments, a (ASW and f/2), b (well-fed and starved). Values are mean ± SE.

**Figure 2 microorganisms-11-00292-f002:**
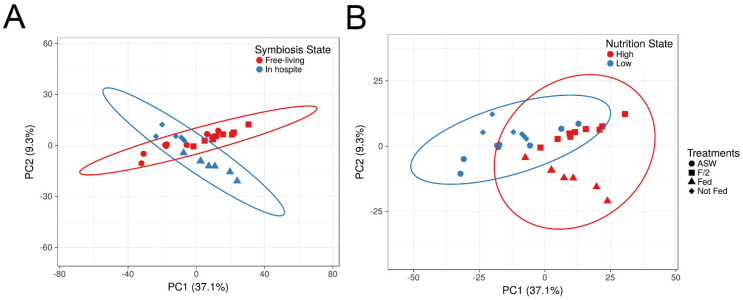
Principal component analysis of all detected *Breviolum minutum* proteins. (**A**) Contrasting algae *in hospite* with the sea anemone Aiptasia versus free-living cells; (**B**) high (i.e., well-fed *in hospite* or cultured in replete f/2 medium) versus low nutritional regimes, meaning starved *in hospite* or cultured in artificial seawater (ASW). Ellipses represent 99% confidence intervals.

**Figure 3 microorganisms-11-00292-f003:**
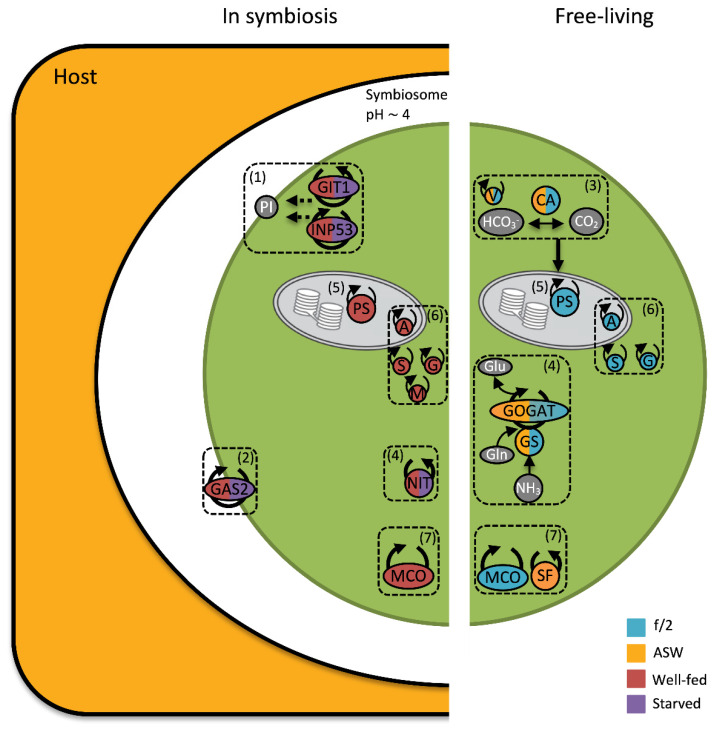
Conceptual diagram of the cellular processes under varying degrees of *B. minutum* compatibility. Coloured proteins were significantly more abundant in the colour-coded treatments. Grey = metabolites/solutes; Purple = Starved; Red = Well-fed; Orange = ASW; Turquoise = f/2. (1) Proteins involved in phosphoinositol (PI) manipulation, including GIT1 and INP53 were more abundant only *in hospite*, potentially for preventing host autophagy, and aiding communication and cell recognition. (2) Cell wall modification proteins (e.g., GAS2) were increased in symbionts regardless of nutritional state. (3) *B. minutum* carbonic anhydrase (CA) and vacuolar proton pumps (V), which work in harmony to convert bicarbonate ions (HCO_3_^-^) to carbon dioxide (CO_2_), were abundant in both free-living treatments. CO_2_ can be further used for photosynthesis. Upregulation of algal CAs and V-Type ATPase in the free-living state might be a response to a lack of readily available CO_2_ provided by the host CAs and proton pumps when *in hospite*. (4) Ammonium is assimilated by *B. minutum* via the GS/GOGAT pathway; however, proteins associated with this pathway increased only in the free-living state, likely because the host controls the flow of nitrogen to its symbionts when *in hospite*. Moreover, nitrilase (NIT), which is involved in hydrolysing nitrile compounds for ammonia, only increased in abundance when *in hospite*, possibly as a response to nitrogen limitation inside the host. (5) Photosystem (PS) I and II proteins involved in light harvesting, as well as chloroplastic electron transport proteins, were highly increased in abundance in the nutrient-enriched treatments (i.e., Well-fed and f/2). (6) Proteins involved in *B. minutum* antioxidant mechanisms such as ascorbate peroxidase (A), superoxide dismutase (S), and glutathione s-transferase (G) were more abundant in high nutrient treatments (i.e., Well-fed and f/2), potentially as a cellular response for maintaining high rates of photosynthesis/respiration. Moreover, metacaspase-1B (M), which is involved in oxidative stress and apoptosis, was only abundant in the well-fed treatment *in hospite*. (7) Nutrient usage, such as for iron, was affected by nutrient regime. Multicopper oxidase (MCO), which is involved in the iron uptake pathway, increased in abundance in the high nutrient regimes. Moreover, soma ferritin (SF), an important iron storage protein, was more abundant in the ASW treatment.

**Table 1 microorganisms-11-00292-t001:** Summary of proteins that are differentially abundant (FDR < 0.05) in *Breviolum minutum* when comparing *in hospite* vs. *ex hospite*, in order of log_2_ fold-change. Proteins shown are those discussed in the text. The full list of proteins that are differentially expressed *in hospite* vs. *ex hospite* is available in Appendix A.

*In* vs. *Ex hospite* Fold Change	Uniprot ID	Uniprot Protein Names
Unique, *in hospite*	A0A1Q9DR34	Polyphosphatidylinositol phosphatase (INP53)
Unique, *in hospite*	A0A1Q9F6G5	Nitrilase
Unique, *in hospite*	Q59Q30	Glycerophosphoinositol permease 1 (GIT1)
Unique, *in hospite*	Q04634	Elongation factor 1-alpha (eEF1)
Unique, *in hospite*	Q06135	1,3-beta-glucanosyltransferase (GAS2)
3.44	B0Y8K2	1,4-beta-D-glucan cellobiohydrolase B (CBHB)
1.78	A0A1Q9CUD0	Putative glucose-6-phosphate 1-epimerase
1.76	Q9SAU2	Ribulose-5-phosphate-3-epimerase (R5P3E)
1.16	B4R9R7	Guanosine monophosphate synthetase (GMP)
−0.92	P11029	Acetyl-CoA carboxylase (ACC)
−1.03	Q9Y7Y7	1,3-beta-glucanosyltransferase (GAS4)
−1.10	Q54WR9	Type-3 glutamine synthetase
−1.49	Q12613	Glutamine synthetase (GS)
−1.50	Q00955	Acetyl-CoA carboxylase (ACC)
−1.56	A0A1Q9CUK2	GDP-mannose transporter GONST5
−2.04	Q9LV03	Glutamate synthase 1, chloroplastic
−3.21	Q76NM6	V-type proton ATPase catalytic subunit A
−3.90	A0A1Q9D9X1	Carbonic anhydrase 2
−3.95	Q9I262	Carbonic anhydrase
Unique, *ex hospite*	P74572	Pyrroline-5-carboxylate reductase (P5CR)
Unique, *ex hospite*	Q9LV03	Glutamate synthase 1, chloroplastic
Unique, *ex hospite*	A5CZ28	Gamma-glutamyl phosphate reductase (GPR)
Unique, *ex hospite*	A0A1Q9CS65	Carbonic anhydrase 2
Unique, *ex hospite*	Q9I262	Carbonic anhydrase
Unique, *ex hospite*	G4N296	Highly reducing polyketide synthase 19 (PKS19)
Unique, *ex hospite*	P9WQE6	Phenolphthiocerol/phthiocerol polyketide synthase (PpsA)
Unique, *ex hospite*	P21567	Alpha-amylase
Unique, *ex hospite*	G2Q9T3	Polysaccharide monooxygenase

**Table 2 microorganisms-11-00292-t002:** Summary of proteins that are differentially abundant (FDR < 0.05) in *Breviolum minutum* when comparing high nutrient vs. low nutrient regimes (i.e., f/2 and well-fed treatments vs. ASW and starved treatments), in order of log_2_ fold-change. Proteins shown here are those discussed in the text. The full list of proteins that were differentially expressed between high vs. low nutrient regimes is available in Appendix A.

High vs. Low Nutrient Regime Fold Change	Uniprot ID	Uniprot Protein Names
Unique, High	A0A1Q9CJD4	Glutathione S-transferase
Unique, High	A0A1Q9CL40	Multicopper oxidase (MCO)
3.54	O09452	Glyceraldehyde-3-phosphate dehydrogenase
3.01	P26302	Phosphoribulokinase (PRK)
2.88	P00110	Cytochrome *c*6 (Cytochrome c-553)
2.87	Q40296	Chlorophyll *a*-chlorophyll *c_2_*-peridinin-protein (apcPC)
2.84	P00110	Cytochrome *c*6 (Cytochrome c-553)
2.78	Q40297	Chlorophyll *a*-chlorophyll *c_2_*-peridinin-protein (apcPC)
2.77	Q95AG0	Cytochrome f
2.76	A0T0C6	Cytochrome *c*-550 (Cytochrome c550)
2.68	Q9XQV2	Photosystem I P700 chlorophyll a apoprotein A2
2.58	Q85FP8	Photosystem I reaction centre subunit XI
2.53	Q9SDM1	Chlorophyll *a-b* binding protein
2.53	Q00598	Ferredoxin--NADP reductase
2.5	P55738	Caroteno-chlorophyll *a-c*-binding protein
2.39	Q41406	Ribulose bisphosphate carboxylase (RuBisCO)
2.35	A0A1Q9EQB8	Metacaspase-1B
2.34	P49472	Photosystem II CP43 reaction centre protein
2.32	P25851	Fructose-1,6-bisphosphatase 1, (FBPase1) (ELECTRON FLOW 1)
2.31	P54375	Superoxide dismutase (SOD)
2.23	A0T0T0	Photosystem II D2 protein
1.69	Q9MSC2	Photosystem II protein D1
1.64	P20136	Glutathione S-transferase 2 (GST-CL2)
1.47	Q7XJ02	Probable L-ascorbate peroxidase 7 (APX)
1.39	P46434	Glutathione S-transferase 1
1.21	P42770	Glutathione reductase, chloroplastic (GR)
0.92	P46429	Glutathione S-transferase 2
−1.07	P42577	Soma ferritin

## Data Availability

Mass spectrometry data are publicly available from the ProteomeXchange Consortium via the PRIDE repository with identifier PXD036981.

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
