# Peer review of "The Influence of Symbiosis on the Proteome of the Exaiptasia Endosymbiont Breviolum minutum"

_microorganisms, 2023, doi:10.3390/microorganisms11020292_

Round 1

Reviewer 1 Report

Article authorised by Mashini et al. "The influence of symbiosis on the proteome of the symbiotic dinoflagellate Breviolum minutum' is a very thorough proteomic analysis of various conditions of symbiotic interactions of Breviolum minutum, a native symbiont of the sea, with anemone Exaiptasia diaphana (‘Aiptasia'). The work was done in accordance with all state-of-the-art requirements for such experiments, the results were processed in an appropriate way and are not subject to doubt. I believe that the article can be accepted for publication, as it is an excellent example of high-quality material that lays the foundation for understanding the molecular mechanisms of symbiotic interactions.

Author Response

Response has been attached as PDF

Reviewer 2 Report

REVIEW OF THE ARTICLE BY AMIRHOSSEIN GHEITANCHI MASHINI ET AL. ENTITLED 'THE INFLUENCE OF SYMBIOSIS ON THE PROTEOME OF THE SYMBIOTIC DINOFLAGELLATE BREVIOLUM MINUTUM'

The authors studied the community of the invertebrate Exaiptasia diaphana (Cnidaria) and the dinophyte Breviolum minutum. They described an effect of interaction with the animal and nutrient status on the protein profile and provide a hypothesis of these changes on regulation of cellular pathways. The work is in scope of the journal. Introduction and methods are sufficient. Results description is sutisfactory. Some changes should be made. There are some conceptual questions to results (see below). Name of the animal should be in the title.

INTRODUCTION

-Please, highlight clearly the novelty.

MATERIALS AND METHODS

-L. 90-97. Please, add avaliable GenBank IDs for sequences.

-L. 118. At least, at first mention, please, use the the full term "maximal photochemical quantum yield".

-L. 119. Please, defind , what is Fv. Were the samples dark-acclimated?

-L. 123. What are "measuring light 4, saturation intensity 4"? Units? Please, indicate type of LED and color (wavelengths).

-Please comemnt on PCA: how did you perform clusterization? There is no evidence for "protein abundance as being distinctly different between symbiotic states and nutritional regimes" (l. 236-237).

RESUTS AND DISCUSSION

-Is there a visualization of interactions between the algae and the 'host' (e.g. microscopy)? Are there specific spation patterns of interactions?

-What do you mean on 'symbiosis' and 'symbiont' in the work? What do this term mean and which kriterion of symbiosis did you use?

-L. 206-207. It is overinterpretation. Comparison of f/2 and ASW (Figure 1B) shows nitrogen level did not affect signifficantly Fv/Fm. It als could be due to P, O, and C level, pH and control of algal metabolism by the 'host' (e.g. by signalling molecules). See, e.g. you own discussion in l. 220.

-Table 1,2. Remove bold and line in the second line. Change alpha and beta to corresponding Greec symbols. Unify 'Photosystem II D2 protein' and  'Photosystem II protein D1'.

-Figure 2. Panels A and B should be cited separatly.

-L. 340-341. The statement is unclear. 'holobiont' is the same as 'the host and its microbial consortia'; dinoflagellates are also part of the microbial consorcium.

-L. 397-399. A terminological confusion. 'Oxidative pentose phosphate pathway' (Warburg–Lipmann–Dickens–Horecker pathway) is also a type of glycolisis. Do you mean 'in parallel with Embden-Meyerhof-Parnas'?

-L. 420-430. Fatty acids are also lipids.

-L. 446, 451. Format of references?

-L. 453. What do you mean? P700 is a chlorophyll dimer in the PS I RC, not a protein.

-L. 457. rubsico should be RuBisCo.

-L. 463. 'quantum yields (Fv/Fm)' change to Fv/Fm.

-L. 464. What is PS?

-L. 472. It is actually should be suoeroxide anion radical (O2−.).

Author Response

Response is attached as PDF
